# Sublinear Time Orthogonal Tensor Decomposition[*]

**Zhao Song**[‡]    **David P. Woodruff**[†]    **Huan Zhang**[⋆]

[‡]Dept. of Computer Science, University of Texas, Austin, USA

[†]IBM Almaden Research Center, San Jose, USA

[⋆]Dept. of Electrical and Computer Engineering, University of California, Davis, USA

zhaos@utexas.edu, dpwoodru@us.ibm.com, ecezhang@ucdavis.edu

## Abstract

A recent work (Wang et. al., NIPS 2015) gives the fastest known algorithms for orthogonal tensor decomposition with provable guarantees. Their algorithm is based on computing sketches of the input tensor, which requires reading the entire input. We show in a number of cases one can achieve the same theoretical guarantees in sublinear time, i.e., even without reading most of the input tensor. Instead of using sketches to estimate inner products in tensor decomposition algorithms, we use importance sampling. To achieve sublinear time, we need to know the norms of tensor slices, and we show how to do this in a number of important cases. For symmetric tensors $\mathbf{T} = \sum_{i=1}^{k} \lambda_i u_i^{\otimes p}$ with $\lambda_i > 0$ for all $i$, we estimate such norms in sublinear time whenever $p$ is even. For the important case of $p = 3$ and small values of $k$, we can also estimate such norms. For asymmetric tensors sublinear time is not possible in general, but we show if the tensor slice norms are just slightly below $\| \mathbf{T} \|_F$ then sublinear time is again possible. One of the main strengths of our work is empirical - in a number of cases our algorithm is orders of magnitude faster than existing methods with the same accuracy.

## 1 Introduction

Tensors are a powerful tool for dealing with multi-modal and multi-relational data. In recommendation systems, often using more than two attributes can lead to better recommendations. This could occur, for example, in Groupon where one could look at users, activities, and time (season, time of day, weekday/weekend, etc.), as three attributes to base predictions on (see [13] for a discussion). Similar to low rank matrix approximation, we seek a tensor decomposition to succinctly store the tensor and to apply it quickly. A popular decomposition method is the canonical polyadic decomposition, i.e., the CANDECOMP/PARAFAC (CP) decomposition, where the tensor is decomposed into a sum of rank-1 components [9]. We refer the reader to [23], where applications of CP including data mining, computational neuroscience, and statistical learning for latent variable models are mentioned.

A natural question, given the emergence of large data sets, is whether such decompositions can be performed quickly. There are a number of works on this topic [17, 16, 7, 11, 10, 4, 20]. Most related to ours are several recent works of Wang et al. [23] and Tung et al. [18], in which it is shown how to significantly speed up this decomposition for orthogonal tensor decomposition using the randomized technique of linear sketching [15]. In this work we also focus on orthogonal tensor decomposition. The idea in [23] is to create a succinct sketch of the input tensor, from which one can then perform implicit tensor decomposition by approximating inner products in existing decomposition methods.

Existing methods, like the power method, involve computing the inner product of a vector, which is now a rank-1 matrix, with another vector, which is now a slice of a tensor. Such inner products can

---

[*]Full version appears on arXiv, 2017. [‡]Work done while visiting IBM Almaden.

[†]Supported by XDATA DARPA Air Force Research Laboratory contract FA8750-12-C-0323.

be approximated much faster by instead computing the inner product of the sketched vectors, which have significantly lower dimension. One can also replace the sketching with sampling to approximate inner products; we discuss some sampling schemes [17, 4] below and compare them to our work.

## 1.1 Our Contributions

We show in a number of important cases, one can achieve the same theoretical guarantees in the work of Wang et al. [23] (which was applied later by Tung et al. [18]), in *sublinear time*, that is, without reading most of the input tensor. While previous work needs to walk through the input at least once to create a sketch, we show one can instead perform *importance sampling* of the tensor based on the current iterate, together with reading a few entries of the tensor which help us learn the norms of tensor slices. We use a version of $\ell_2$-sampling for our importance sampling. One source of speedup in our work and in Wang et al. [23] comes from approximating inner products in iterations in the robust tensor power method (see below). To estimate $\langle u, v \rangle$ for $n$-dimensional vectors $u$ and $v$, their work computes sketches $S(u)$ and $S(v)$ and approximates $\langle u, v \rangle \approx \langle S(u), S(v) \rangle$. Instead, if one has $u$, one can sample coordinates $i$ proportional to $u_i^2$, which is known as $\ell_2$-sampling [14, 8]. One estimates $\langle u, v \rangle$ as $\frac{v_i \|u\|_2^2}{u_i}$, which is unbiased and has variance $O(\|u\|_2^2 \|v\|_2^2)$. These guarantees are similar to those using sketching, though the constants are significantly smaller (see below), and unlike sketching, one does not need to read the entire tensor to perform such sampling.

**Symmetric Tensors:** As in [23], we focus on orthogonal tensor decomposition of symmetric tensors, though we explain the extension to the asymmetric case below. Symmetric tensors arise in engineering applications, for example, to represent the symmetric tensor field of stress, strain, and anisotropic conductivity. Another example is diffusion MRI in which one uses symmetric tensors to describe diffusion in the brain or other parts of the body. In spectral methods symmetric tensors are exactly those that come up in Latent Dirichlet Allocation problems. Although one can symmetrize a tensor using simple matrix operations (see, e.g., [1]), we cannot do this in sublinear time.

In orthogonal tensor decompostion of a symmetric matrix, there is an underlying $n \times n \cdots n$ tensor $\mathbf{T}^* = \sum_{i=1}^k \lambda_i v_i^{\otimes p}$, and the input tensor is $\mathbf{T} = \mathbf{T}^* + \mathbf{E}$, where $\|\mathbf{E}\|_2 \leq \epsilon$. We have $\lambda_1 > \lambda_2 > \cdots > \lambda_k > 0$ and that $\{v_i\}_{i=1}^k$ is a set of orthonormal vectors. The goal is to reconstruct approximations $\hat{v}_i$ to the vectors $v_i$, and approximations $\hat{\lambda}_i$ to the $\lambda_i$. Our results naturally generalize to tensors with different lengths in different dimensions. For simplicity, we first focus on order $p = 3$.

In the robust tensor power method [1], one generates a random initial vector $u$, and performs $T$ update steps $\hat{u} = \mathbf{T}(I, u, u)/\|\mathbf{T}(I, u, u)\|_2$, where

$$\mathbf{T}(I, u, u) = \Big[\sum_{j=1}^n \sum_{\ell=1}^n \mathbf{T}_{1,j,\ell} \, u_j u_\ell, \sum_{j=1}^n \sum_{\ell=1}^n \mathbf{T}_{2,j,\ell} \, u_j u_\ell, \cdots, \sum_{j=1}^n \sum_{\ell=1}^n \mathbf{T}_{n,j,\ell} \, u_j u_\ell\Big].$$

The matrices $\mathbf{T}_{1,*,*}, \ldots, \mathbf{T}_{n,*,*}$ are referred to as the slices. The vector $\hat{u}$ typically converges to the top eigenvector in a small number of iterations, and one often chooses a small number $L$ of random initial vectors to boost confidence. Successive eigenvectors can be found by deflation. The algorithm and analysis immediately extend to higher order tensors.

We use $\ell_2$-sampling to estimate $\mathbf{T}(I, u, u)$. To achieve the same guarantees as in [23], for typical settings of parameters (constant $k$ and several eigenvalue assumptions) naïvely one needs to take $O(n^2)$ $\ell_2$-samples from $u$ *for each slice* in each iteration, resulting in $\Omega(n^3)$ time and destroying our sublinearity. We observe that if we additionally knew the squared norms $\|\mathbf{T}_{1,*,*}\|_F^2, \ldots, \|\mathbf{T}_{n,*,*}\|_F^2$, then we could take $O(n^2)$ $\ell_2$-samples *in total*, where we take $\frac{\|\mathbf{T}_{i,*,*}\|_F^2}{\|\mathbf{T}\|_F^2} \cdot O(n^2)$ $\ell_2$-samples from the $i$-th slice in expectation. Perhaps in some applications such norms are known or cheap to compute in a single pass, but without further assumptions, how can one obtain such norms in sublinear time?

If $\mathbf{T}$ is a symmetric tensor, then $\mathbf{T}_{j,j,j} = \sum_{i=1}^k \lambda_i v_{i,j}^3 + \mathbf{E}_{j,j,j}$. Note that if there were no noise, then we could read off approximations to the slice norms, since $\|\mathbf{T}_{j,*,*}\|_F^2 = \sum_{i=1}^k \lambda_i^2 v_{i,j}^2$, and so $\mathbf{T}_{j,j,j}^{2/3}$ is an approximation to $\|\mathbf{T}_{j,*,*}\|_F^2$ up to factors depending on $k$ and the eigenvalues. However, there is indeed noise. To obtain non-trivial guarantees, the robust tensor power method assumes $\|\mathbf{E}\|_2 = O(1/n)$, where

$$\|\mathbf{E}\|_2 = \sup_{\|u\|_2 = \|v\|_2 = \|w\|_2 = 1} \mathbf{E}(u, v, w) = \sup_{\|u\|_2 = \|v\|_2 = \|w\|_2 = 1} \sum_{i=1}^n \sum_{j=1}^n \sum_{k=1}^n \mathbf{E}_{i,j,k} \, u_i v_j w_k,$$

which in particular implies $|\mathbf{E}_{j,j,j}| = O(1/n)$. This assumption comes from the $\Theta(1/\sqrt{n})$-correlation of the random initial vector to $v_1$. This noise bound does not trivialize the problem; indeed, $\mathbf{E}_{j,j,j}$ can be chosen adversarially subject to $|\mathbf{E}_{j,j,j}| = O(1/n)$, and if the $v_i$ were random unit vectors and the $\lambda_i$ and $k$ were constant, then $\sum_{i=1}^{k} \lambda_i v_{i,j}^3 = O(1/n^{3/2})$, which is small enough to be completely masked by the noise $\mathbf{E}_{j,j,j}$. Nevertheless, there is a lot of information about the slice norms. Indeed, suppose $k = 1$, $\lambda_1 = \Theta(1)$, and $\|\mathbf{T}\|_F = 1$. Then $\mathbf{T}_{j,j,j} = \Theta(v_{1,j}^3) + \mathbf{E}_{j,j,j}$, and one can show $\|\mathbf{T}_{j,*,*}\|_F^2 = \lambda_1^2 v_{1,j}^2 \pm O(1/n)$. Again using that $|\mathbf{E}_{j,j,j}| = O(1/n)$, this implies $\|\mathbf{T}_{j,*,*}\|_F^2 = \omega(n^{-2/3})$ if and only if $\mathbf{T}_{j,j,j} = \omega(1/n)$, and therefore one would notice this by reading $\mathbf{T}_{j,j,j}$. There can only be $o(n^{2/3})$ slices $j$ for which $\|\mathbf{T}_{j,*,*}\|_F^2 = \omega(n^{-2/3})$, since $\|\mathbf{T}\|_F^2 = 1$. Therefore, for each of them we can afford to take $O(n^2)$ $\ell_2$-samples and still have an $O(n^{2+2/3}) = o(n^3)$ sublinear running time. The remaining slices all have $\|\mathbf{T}_{j,*,*}\|_F^2 = O(n^{-2/3})$, and therefore if we also take $O(n^{1/3})$ $\ell_2$-samples from *every slice*, we will also estimate the contribution to $\mathbf{T}(I, u, u)$ from these slices well. This is also a sublinear $O(n^{2+1/3})$ number of samples.

While the previous paragraph illustrates the idea for $k = 1$, for $k = 2$ we need to read more than the $\mathbf{T}_{j,j,j}$ entries to decide how many $\ell_2$-samples to take from a slice. The analysis is more complicated because of *sign cancellations*. Even for $k = 2$ we could have $\mathbf{T}_{j,j,j} = \lambda_1 v_{1,j}^3 + \lambda_2 v_{2,j}^3 + \mathbf{E}_{j,j,j}$, and if $v_{1,j} = -v_{2,j}$ then we may not detect that $\|\mathbf{T}_{j,*,*}\|_F^2$ is large. We fix this by also reading the entries $\mathbf{T}_{i,j,j}$, $\mathbf{T}_{j,i,j}$, and $\mathbf{T}_{j,j,i}$ for every $i$ and $j$. This is still only $O(n^2)$ entries and so we are still sublinear time. Without additional assumptions, we only give a formal analysis of this for $k \in \{1, 2\}$.

More importantly, if instead of third-order symmetric tensors we consider $p$-th order symmetric tensors for even $p$, we do not have such sign cancellations. In this case we do not have any restrictions on $k$ for estimating slice norms. One does need to show after deflation, the slice norms can still be estimated; this holds because the eigenvectors and eigenvalues are estimated sufficiently well.

We also give several per-iteration optimizations of our algorithm, based on careful implementations of generating a sorted list of random numbers and random permutations. We find empirically (see below) that we are much faster per iteration than previous sketching algorithms, in addition to not having to read the entire input tensor in a preprocessing step.

**Asymmetric Tensors:** For asymmetric tensors, e.g., 3rd-order tensors of the form $\sum_{i=1}^{k} \lambda_i u_i \otimes v_i \otimes w_i$, it is impossible to achieve sublinear time in general, since it is hard to distinguish $\mathbf{T} = e_i \otimes e_j \otimes e_k$ for random $i, j, k \in \{1, 2, \ldots, n\}$ from $\mathbf{T} = 0^{\otimes 3}$. We make a necessary and sufficient assumption that all the entries of the $u_i$ are less than $n^{-\gamma}$ for an arbitrarily small constant $\gamma > 0$. In this case, all slice norms are $o(n^{-\gamma})$ and by taking $O(n^{2-\gamma})$ samples from each slice we achieve sublinear time. We can also apply such an assumption to symmetric tensors.

**Empirical Results:** One of the main strengths of our work is our empirical results. In each iteration we approximate $\mathbf{T}(I, u, u)$ a total of $B$ times independently and take the median to increase our confidence. In the notation of [23], $B$ corresponds to the number of independent sketches used. While the median works empirically, there are some theoretical issues with it discussed in Remark 4. Also let $b$ be the total number of $\ell_2$-samples we take per iteration, which corresponds to the sketch size in the notation of [23]. We found that empirically we can set $B$ and $b$ to be much smaller than that in [23] and achieve the same error guarantees. One explanation for this is that the variance bound we obtain via importance sampling is a factor of $4^3 = 64$ smaller than in [23], and for $p$-th order tensors, a factor of $4^p$ smaller.

To give an idea of how much smaller we can set $b$ and $B$, to achieve roughly the same squared residual norm error on the synthetic data sets of dimension 1200 for finding a good rank-1 approximation, the algorithm of [23] would need to set parameters $b = 2^{16}$ and $B = 50$, whereas we can set $b = 10 \times 1200$ and $B = 5$. Our running time is 2.595 seconds and we have no preprocessing time, whereas the algorithm of [23] has a running time of 116.3 seconds and 55.34 seconds of preprocessing time. We refer the reader to Table 1 in Section 3. In total we are over 50 times faster.

We also demonstrate our algorithm in a real-world application using real datasets, even when the datasets are sparse. Namely, we consider a spectral algorithm for Latent Dirichlet Allocation [1, 2] which uses tensor decomposition as its core computational step. We show a significant speedup can be achieved on tensors occurring in applications such as LDA, and we refer the reader to Table 2 in

Section 3. For example, on the wiki [23] dataset with a tensor dimension of 200, we run more than 5 times faster than the sketching-based method.

**Previous Sampling Algorithms:** Previous sampling-based schemes of [17, 4] do not achieve our guarantees, because [17] uses uniform sampling, which does not work for tensors with spiky elements, while the non-uniform sampling in [4] requires touching all of the entries in the tensor and making two passes over it.

**Notation** Let $[n]$ denote $\{1, 2, \ldots, n\}$. Let $\otimes$ denote the outer product, and $u^{\otimes 3} = u \otimes u \otimes u$. Let $\mathbf{T} \in \mathbb{R}^{n^p}$, where $p$ is the order of tensor $\mathbf{T}$ and $n$ is the dimension of tensor $\mathbf{T}$. Let $\langle \mathbf{A}, \mathbf{B} \rangle$ denote the entry-wise inner product between two tensors $\mathbf{A}, \mathbf{B} \in \mathbb{R}^{n^p}$, e.g., $\langle \mathbf{A}, \mathbf{B} \rangle = \sum_{i_1=1}^{n} \sum_{i_2=1}^{n} \cdots \sum_{i_p=1}^{n} \mathbf{A}_{i_1, i_2, \cdots, i_p} \cdot \mathbf{B}_{i_1, i_2, \cdots, i_p}$. For a tensor $\mathbf{A} \in \mathbb{R}^{n^p}$, $\| \mathbf{A} \|_F = (\sum_{i_1=1}^{n} \sum_{i_2=1}^{n} \cdots \sum_{i_p=1}^{n} \mathbf{A}_{i_1, \cdots, i_p}^2)^{\frac{1}{2}}$. For random variable $X$ let $\mathbb{E}[X]$ denote its expectation of $X$ and $\mathbb{V}[X]$ its variance (if these quantities exist).

## 2 Main Results

We explain the details of our main results in this section. First, we state the importance sampling lemmas for our tensor application. Second, we explain how to quickly produce a list of random tuples according to a certain distribution needed by our algorithm. Third, we combine the first and the second parts to get a fast way of approximating tensor contractions, which are used as subroutines in each iteration of the robust tensor power method. We then provide our main theoretical results, and how to estimate the slice norms needed by our main algorithm.

**Importance sampling lemmas.** Approximating an inner product is a simple application of importance sampling. Tensor contraction $\mathbf{T}(u, v, w)$ can be regarded as the inner product between two $n^3$-dimensional vectors, and thus importance sampling can be applied. Lemma 1 suggests that we can take a few samples according to their importance, e.g., we can sample $\mathbf{T}_{i,j,k} u_i v_j w_k$ with probability $|u_i v_j w_k|^2 / \|u\|_2^2 \|v\|_2^2 \|w\|_2^2$. As long as the number of samples is large enough, it will approximate the true tensor contraction $\sum_i \sum_j \sum_k \mathbf{T}_{i,j,k} u_i v_j w_k$ with small variance after a final rescaling.

**Lemma 1.** *Suppose random variable $X = \mathbf{T}_{i,j,k} u_i v_j w_k / (p_i q_j r_k)$ with probability $p_i q_j r_k$ where $p_i = |u_i|^2 / \|u\|_2^2$, $q_j = |v_j|^2 / \|v\|_2^2$, and $r_k = |w_k|^2 / \|w\|_2^2$, and we take $L$ i.i.d. samples of $X$, denoted $X_1, X_2, \cdots, X_L$. Let $Y = \frac{1}{L} \sum_{\ell=1}^{L} X_\ell$. Then (1) $\mathbb{E}[Y] = \langle \mathbf{T}, u \otimes v \otimes w \rangle$, and (2) $\mathbb{V}[Y] \le \frac{1}{L} \| \mathbf{T} \|_F^2 \cdot \|u \otimes v \otimes w\|_F^2$.*

Similarly, we also have importance sampling for each slice $\mathbf{T}_{i,*,*}$, i.e., "face" of $\mathbf{T}$.

**Lemma 2.** *For all $i \in [n]$, suppose random variable $X^i = \mathbf{T}_{i,j,k} v_j w_k / (q_j r_k)$ with probability $q_j r_k$, where $q_j = |v_j|^2 / \|v\|_2^2$ and $r_k = |w_k|^2 / \|w\|_2^2$, and we take $L_i$ i.i.d. samples of $X^i$, say $X_1^i, X_2^i, \cdots, X_{L_i}^i$. Let $Y^i = \frac{1}{L_i} \sum_{\ell=1}^{L} X_\ell^i$. Then (1) $\mathbb{E}[Y^i] = \langle \mathbf{T}_{i,*,*}, v \otimes w \rangle$ and (2) $\mathbb{V}[Y^i] \le \frac{1}{L_i} \| \mathbf{T}_{i,*,*} \|_F^2 \|v \otimes w\|_F^2$.*

**Generating importance samples in linear time.** We need an efficient way to sample indices of a vector based on their importance. We view this problem as follows: imagine $[0, 1]$ is divided into $z$ "bins" with different lengths corresponding to the probability of selecting each bin, where $z$ is the number of indices in a probability vector. We generate $m$ random numbers uniformly from $[0, 1]$ and see which bin each random number belongs to. If a random number is in bin $i$, we sample the $i$-th index of a vector. There are known algorithms [6, 19] to solve this problem in $O(z + m)$ time.

We give an alternative algorithm GENRANDTUPLES. Our algorithm combines Bentley and Saxe's algorithm [3] for efficiently generating $m$ sorted random numbers in $O(m)$ time, and Knuth's shuffling algorithm [12] for generating a random permutation of $[m]$ in $O(m)$ time. We use the notation CUMPROB$(v, w)$ and CUMPROB$(u, v, w)$ for the algorithm creating the distributions on $\mathbb{R}^{n^2}$ and $\mathbb{R}^{n^3}$ of Lemma 2 and Lemma 1, respectively. We note that naïvely applying previous algorithms would require $z = O(n^2)$ and $z = O(n^3)$ time to form these two distributions, but we can take $O(m)$ samples from them implicitly in $O(n + m)$ time.

**Fast approximate tensor contractions.** We propose a fast way to approximately compute tensor contractions $\mathbf{T}(I, v, w)$ and $\mathbf{T}(u, v, w)$ with a sublinear number of samples of $\mathbf{T}$, as shown in Alogrithm 1 and Algorithm 2. Naïvely computing tensor contractions using all of the entries of $\mathbf{T}$ gives an exact answer but could take $n^3$ time. Also, to keep our algorithm sublinear time, we never explicitly compute the deflated tensor; rather we represent it implicitly and sample from it.

**Algorithm 1** Subroutine for approximate tensor contraction $\mathbf{T}(I, v, w)$

1: **function** APPROXTIVW($\mathbf{T}, v, w, n, B, \{\widehat{b}_i\}$)
2:   $\widetilde{q}, \widetilde{r} \leftarrow$ CUMPROB($v, w$)
3:   **for** $d = 1 \to B$ **do**
4:     $\mathcal{L} \leftarrow$ GENRANDTUPLES($\sum_{i=1}^{n} \widehat{b}_i, \widetilde{q}, \widetilde{r}$)
5:     **for** $i = 1 \to n$ **do**
6:       $s_i^{(d)} \leftarrow 0$
7:       **for** $\ell = 1 \to \widehat{b}_i$ **do**
8:         $(j, k) \leftarrow \mathcal{L}_{(i-1)b+\ell}$
9:         $s_i^{(d)} \leftarrow s_i^{(d)} + \frac{1}{q_j r_k} \mathbf{T}_{i,j,k} \cdot u_j \cdot u_k$
10:   $\widehat{\mathbf{T}}(I, v, w)_i \leftarrow \underset{d \in [B]}{\text{median}}\ s_i^{(d)} / \widehat{b}_i, \forall i \in [n]$

11: **return** $\widehat{\mathbf{T}}(I, v, w)$

**Algorithm 2** Subroutine for approximate tensor contraction $\mathbf{T}(u, v, w)$

1: **function** APPROXTUVW($\mathbf{T}, u, v, w, n, B, \widehat{b}$)
2:   $\widetilde{p}, \widetilde{q}, \widetilde{r} \leftarrow$ CUMPROB($u, v, w$)
3:   **for** $d = 1 \to B$ **do**
4:     $\mathcal{L} \leftarrow$ GENRANDTUPLES($\widehat{b}, \widetilde{p}, \widetilde{q}, \widetilde{r}$).
5:     $s^{(d)} \leftarrow 0$
6:     **for** $(i, j, k) \in \mathcal{L}$ **do**
7:       $s^{(d)} \leftarrow s^{(d)} + \frac{1}{p_i q_j r_k} \mathbf{T}_{i,j,k} \cdot u_i \cdot u_j \cdot u_k$
8:     $s^{(d)} \leftarrow s^{(d)} / \widehat{b}$
9:   $\widehat{\mathbf{T}}(u, v, w) \leftarrow \underset{d \in [B]}{\text{median}}\ s^{(d)}$

10: **return** $\widehat{\mathbf{T}}(u, v, w)$

The following theorem gives the error bounds of APPROXTIVW and APPROXTUVW (in Algorithm 1 and 2). Let $\widehat{b}_i$ be the number samples we take from slice $i \in [n]$ in APPROXTIVW, and let $\widehat{b}$ denote the total number of samples in our algorithm.

**Theorem 3.** *For* $\mathbf{T} \in \mathbb{R}^{n \times n \times n}$ *and* $u \in \mathbb{R}^n$ *with* $\|u\|_2 = 1$, *define the number* $\varepsilon_{1,\mathbf{T}}(u) = \widehat{\mathbf{T}}(u, u, u) - \mathbf{T}(u, u, u)$ *and the vector* $\varepsilon_{2,\mathbf{T}}(u) = \widehat{\mathbf{T}}(I, u, u) - \mathbf{T}(I, u, u)$. *For any* $b > 0$, *if* $\widehat{b}_i \gtrsim b \| \mathbf{T}_{i,*,*} \|_F^2 / \| \mathbf{T} \|_F^2$ *then the following bounds hold [1]:*

$$\mathbb{E}[|\varepsilon_{1,\mathbf{T}}(u)|^2] = O(\| \mathbf{T} \|_F^2 / b), \text{ and } \mathbb{E}[\|\varepsilon_{2,\mathbf{T}}(u)\|_2^2] = O(n \| \mathbf{T} \|_F^2 / b).$$

*In addition, for any fixed* $\omega \in \mathbb{R}^n$ *with* $\|\omega\|_2 = 1$,

$$\mathbb{E}[\langle \omega, \varepsilon_{2,T}(u) \rangle^2] = O(\| \mathbf{T} \|_F^2 / b). \tag{1}$$

Eq. (1) can be obtained by observing that each random variable $[\varepsilon_{2,\mathbf{T}}(u)]_i$ is independent and so $\mathbb{V}[\langle \omega, \varepsilon_{2,\mathbf{T}}(u) \rangle] = \sum_{i=1}^{n} \omega_i^2 \frac{\| \mathbf{T}_{i,*,*} \|_F^2}{\widehat{b}_i} \lesssim (\sum_{i=1}^{n} \omega_i^2) \frac{\| \mathbf{T} \|_F^2}{b} = \frac{\| \mathbf{T} \|_F^2}{b}$.

**Remark 4.** *In [23], the coordinate-wise median of B estimates to the* $\mathbf{T}(I, v, w)$ *is used to boost the success probability. There appears to be a gap [21] in their argument as it is unclear how to achieve (1) after taking a coordinate-wise median, which is (7) in Theorem 1 of [23]. To fix this, we instead pay a factor proportional to the number of iterations in Algorithm 3 in the sample complexity* $\widehat{b}$. *Since we have expectation bounds on the quantities in Theorem 3, we can apply a Markov bound and a union bound across all iterations. This suffices for our main theorem concerning sublinear time below. One can obtain high probability bounds by running Algorithm 3 multiple times independently, and taking coordinate-wise medians of the output eigenvectors. Empirically, our algorithm works even if we take the median in each iteration, which is done in line 10 in Algorithm 1.*

Replacing Theorem 1 in [23] by our Theorem 3, the rest of the analysis in [23] is unchanged. Our Algorithm 3 is the same as the sketching-based robust tensor power method in [23], except for lines 10, 12, 15, and 17, where the sketching-based approximate tensor contraction is replaced by our importance sampling procedures APPROXTUVW and APPROXTIVW. Rather than use Theorem 2 of Wang et al. [23], the main theorem concerning the correctness of the robust tensor decomposition algorithm, we use a recent improvement of it by Wang and Anandkumar in Theorems 4.1 and 4.2 of [22], which states general guarantees for any algorithm satisfying per iteration noise guarantees. These theorems also remove many of the earlier eigenvalue assumptions in Theorem 2 of [23].

**Theorem 5.** *(Theorem 4.1 and 4.2 of [22]), Suppose* $\mathbf{T} = \mathbf{T}^* + \mathbf{E}$, *where* $\mathbf{T} = \sum_{i=1}^{k} \lambda_i v_i^{\otimes 3}$ *with* $\lambda_i > 0$ *and orthonormal basis vectors* $\{v_1, \ldots, v_k\} \subseteq \mathbb{R}^n$, $n \geq k$. *Let* $\lambda_{\max}, \lambda_{\min}$ *be the largest and smallest values in* $\{\lambda_i\}_{i=1}^{k}$ *and* $\{\widehat{\lambda}_i, \widehat{v}_i\}_{i=1}^{k}$ *be outputs of the robust tensor power method. There exist absolute constants* $K_0, C_0, C_1, C_2, C_3 > 0$ *such that if* $\mathbf{E}$ *satisfies*

$$\| \mathbf{E}(I, u_t^{(\tau)}, u_t^{(\tau)}) \|_2 \leq \epsilon, \quad | \mathbf{E}(v_i, u_t^{(\tau)}, u_t^{(\tau)}) | \leq \min\{\epsilon/\sqrt{k}, C_0 \lambda_{\min}/n\}, \tag{2}$$

**Algorithm 3** Our main algorithm

1: **function** IMPORTANCESAMPLINGRB($\mathbf{T}, n, B, b$)
2:   **if** $s_i$ are known, where $\| \mathbf{T}_{i,*,*} \|_F^2 \lesssim s_i$ **then**
3:     $\widehat{b}_i \leftarrow b \cdot s_i / \| \mathbf{T} \|_F^2, \forall i \in [n]$
4:   **else**
5:     $\widehat{b}_i \leftarrow b/n, \forall i \in [n]$
6:   $\widehat{b} = \sum_{i=1}^{n} \widehat{b}_i$
7:   **for** $\ell = 1 \to L$ **do**
8:     $u^{(\ell)} \leftarrow$ INITIALIZATION
9:     **for** $t = 1 \to T$ **do**
10:       $u^{(\ell)} \leftarrow$ APPROXTIVW($\mathbf{T}, u^{(\ell)}, u^{(\ell)}, n, B, \{\widehat{b}_i\}$)
11:       $u^{(\ell)} \leftarrow u^{(\ell)}/\|u^{(\ell)}\|_2$
12:     $\lambda^{(\ell)} \leftarrow$ APPROXTUVW($\mathbf{T}, u^{(\ell)}, u^{(\ell)}, u^{(\ell)}, n, B, \widehat{b}$)
13:   $\ell^* \leftarrow \arg\max_{\ell \in [L]} \lambda^{(\ell)}, u^* \leftarrow u^{(\ell^*)}$
14:   **for** $t = 1 \to T$ **do**
15:     $u^* \leftarrow$ APPROXTIVW($\mathbf{T}, u^*, u^*, n, B, \{\widehat{b}_i\}$)
16:     $u^* \leftarrow u^*/\|u^*\|_2$
17:   $\lambda^* \leftarrow$ APPROXTUVW($\mathbf{T}, u^*, u^*, u^*, n, B, \widehat{b}$)
18:   **return** $\lambda^*, u^*$

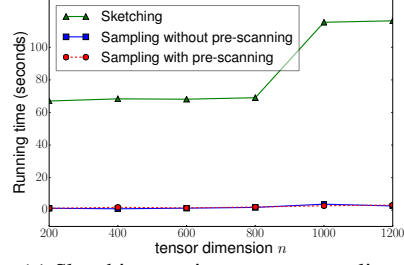
(a) Sketching v.s. importance sampling

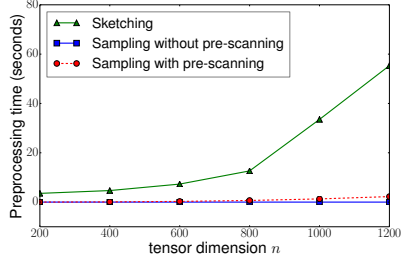
(b) Preprocessing time
Figure 1: Running time with growing dimension

*for all $i \in [k]$, $t \in [T]$, and $\tau \in [L]$ and furthermore*

$$\epsilon \leq C_1 \cdot \lambda_{\min}/\sqrt{k}, \quad T = \Omega(\log(\lambda_{\max} n/\epsilon)), \quad L \geq \max\{K_0, k\} \log(\max\{K_0, k\}),$$

*then with probability at least $9/10$, there exists a permutation $\pi : [k] \to [k]$ such that*

$$|\lambda_i - \widehat{\lambda}_{\pi(i)}| \leq C_2\epsilon, \quad \|v_i - \widehat{v}_{\pi(i)}\|_2 \leq C_3\epsilon/\lambda_i, \quad \forall i = 1, \cdots, k.$$

Combining the previous theorem with our importance sampling analysis, we obtain:

**Theorem 6** (Main). *Assume the notation of Theorem 5. For each $j \in [k]$, suppose we take $\widehat{b}^{(j)} = \sum_{i=1}^{n} \widehat{b}_i^{(j)}$ samples during the power iterations for recovering $\widehat{\lambda}_j$ and $\widehat{v}_j$, the number of samples for slice $i$ is $\widehat{b}_i^{(j)} \gtrsim bkT\|[\mathbf{T} - \sum_{l=1}^{j-1} \widehat{\lambda}_l \widehat{v}_l^{\otimes 3}]_{i,*,*}\|_F^2 / \| \mathbf{T} - \sum_{l=1}^{j-1} \widehat{\lambda}_l \widehat{v}_l^{\otimes 3} \|_F^2$ where $b \gtrsim n\| \mathbf{T} \|_F^2/\epsilon^2 + \| \mathbf{T} \|_F^2/\min\{\epsilon/\sqrt{k}, \lambda_{\min}/n\}^2$. Then the output guarantees of Theorem 5 hold for Algorithm 3 with constant probability. Our total time is $O(LTk^2\widehat{b})$ and the space is $O(nk)$, where $\widehat{b} = \max_{j \in [k]} \widehat{b}^{(j)}$.*

In Theorem 3, if we require $\widehat{b}_i = b\| \mathbf{T}_{i,*,*} \|_F^2/\| \mathbf{T} \|_F^2$, we need to scan the entire tensor to compute $\| \mathbf{T}_{i,*,*} \|_F^2$, making our algorithm not sublinear. With the following mild assumption in Theorem 7, our algorithm is sublinear when sampling uniformly ($\widehat{b}_i = b/n$) without computing $\| \mathbf{T}_{i,*,*} \|_F^2$:

**Theorem 7** (Bounded slice norm). *There is a constant $\alpha > 0$, a constant $\beta \in (0, 1]$ and a sufficiently small constant $\gamma > 0$, such that, for any 3rd order tensor $\mathbf{T} = \mathbf{T}^* + \mathbf{E} \in \mathbb{R}^{n^3}$ with $\text{rank}(\mathbf{T}^*) \leq n^\gamma$, $\lambda_k \geq 1/n^\gamma$, if $\| \mathbf{T}_{i,*,*} \|_F^2 \leq \frac{1}{n^\beta}\| \mathbf{T} \|_F^2$ for all $i \in [n]$, and $\mathbf{E}$ satisfies (2), then Algorithm 3 runs in $O(n^{3-\alpha})$ time.*

The condition $\beta \in (0, 1]$ is a practical one. When $\beta = 1$, all tensor slices have equal Frobenius norm. The case $\beta = 0$ only occurs when $\| \mathbf{T}_{i,*,*} \|_F = \| \mathbf{T} \|_F$; i.e., all except one slice is zero. This theorem can also be applied to asymmetric tensors, since the analysis in [23] can be extended to them.

For certain cases, we can remove the bounded slice norm assumption. The idea is to take a sublinear number of samples from the tensor to obtain upper bounds on all slice norms. In the full version, we extend the algorithm and analysis of the robust tensor power method to $p > 3$ by replacing contractions $\mathbf{T}(u, v, w)$ and $\mathbf{T}(I, v, w)$ with $\mathbf{T}(u_1, u_2, \cdots, u_p)$ and $\mathbf{T}(I, u_2, \cdots, u_p)$. As outlined in Section 1, when $p$ is even, because we do not have sign cancellations we can show:

**Theorem 8** (Even order). *There is a constant $\alpha > 0$ and a sufficiently small constant $\gamma > 0$, such that, for any even order-$p$ tensor $\mathbf{T} = \mathbf{T}^* + \mathbf{E} \in \mathbb{R}^{n^p}$ with $\text{rank}(\mathbf{T}^*) \leq n^\gamma$, $p \leq n^\gamma$ and $\lambda_k \geq 1/n^\gamma$. For any sufficiently large constant $c_0$, there exists a sufficiently small constant $c > 0$, for any $\epsilon \in (0, c\lambda_k/(c_0p^2kn^{(p-2)/2}))$ if $\mathbf{E}$ satisfies $\| \mathbf{E} \|_2 \leq \epsilon/(c_0\sqrt{n})$, Algorithm 3 runs in $O(n^{p-\alpha})$ time.*

As outlined in Section 1, for $p = 3$ and small $k$ we can take sign considerations into account:

**Theorem 9** (Low rank). *There is a constant $\alpha > 0$ and a sufficiently small constant $\gamma > 0$ such that for any symmetric tensor $\mathbf{T} = \mathbf{T}^* + \mathbf{E} \in \mathbb{R}^{n^3}$ with $\mathbf{E}$ satisfying (2), $\text{rank}(\mathbf{T}^*) \leq 2$, and $\lambda_k \geq 1/n^\gamma$, then Algorithm 3 runs in $O(n^{3-\alpha})$ time.*

## 3 Experiments

### 3.1 Experiment Setup and Datasets

Our implementation shares the same code base [1] as the sketching-based robust tensor power method proposed in [23]. We ran our experiments on an i7-5820k CPU with 64 GB of memory in single-threaded mode. We ran two versions of our algorithm: the version *with pre-scanning* scans the full tensor to accurately measure per-slice Frobenius norms and make samples for each slice in proportion to its Frobenius norm in APPROXTIVW; the version *without pre-scanning* assumes that the Frobenius norm of each slice is bounded by $\frac{1}{n^\alpha}\|\mathbf{T}\|_F^2, \alpha \in (0, 1]$ and uses $b/n$ samples per slice, where $b$ is the total number of samples our algorithm makes, analogous to the sketch length $b$ in [23].

**Synthetic datasets.** We first generated an orthonormal basis $\{\boldsymbol{v}_i\}_{i=1}^k$ and then computed the synthetic tensor as $\mathbf{T}^* = \sum_{i=1}^k \lambda_i \boldsymbol{v}_i^{\otimes 3}$, with $\lambda_1 \geq \cdots \geq \lambda_k$. Then we normalized $\mathbf{T}^*$ such that $\|\mathbf{T}^*\|_F = 1$, and added a symmetric Gaussian noise tensor $\mathbf{E}$ where $\mathbf{E}_{ijl} \sim \mathcal{N}(0, \frac{\sigma}{n^{1.5}})$ for $i \leq j \leq l$. Then $\sigma$ controls the noise-to-signal ratio and we kept it as 0.01 in all our synthetic tensors. For the eigenvalues $\lambda_i$, we generated three different decays: inverse decay $\lambda_i = \frac{1}{i}$, inverse square decay $\lambda_i = \frac{1}{i^2}$, and linear decay $\lambda_i = 1 - \frac{i-1}{k}$. We also set $k = 100$ when generating tensors, since higher rank eigenvalues were almost indistinguishable from the added noise. To show the scalability of our algorithm, we generated tensors with different dimensions: $n = 200, 400, 600, 800, 1000, 1200$.

**Real-life datasets.** Latent Dirichlet Allocation [5] (LDA) is a powerful generative statistical model for topic modeling. A spectral method has been proposed to solve LDA models [1, 2] and the most critical step in spectral LDA is to decompose a symmetric $K \times K \times K$ tensor with orthogonal eigenvectors, where $K$ is the number of modeled topics. We followed the steps in [1, 18] and built a $K \times K \times K$ tensor $\mathbf{T}_{\text{LDA}}$ for each dataset, and then ran our algorithms directly on $\mathbf{T}_{\text{LDA}}$ to see how it works on those tensors in real applications. In our experiments we keep $K = 200$. We used the two same datasets as the previous work [23]: Wiki and Enron, as well as four additional real-life datasets. We refer the reader to our GitHub repository [2] for our code and full results.

### 3.2 Results

We considered running time and the squared residual norm to evaluate the performance of our algorithms. Given a tensor $\mathbf{T} \in \mathbb{R}^{n^3}$, let $\|\mathbf{T} - \sum_{i=1}^k \lambda_i u_i \otimes v_i \otimes w_i\|_F^2$ denote the squared residual norm where $\{(\lambda_1, u_1, v_1, w_1), \cdots, (\lambda_k, u_k, v_k, w_k)\}$ are the eigenvalue/eigenvectors obtained by the robust power method. To reduce the experiment time we looked only for the first eigenvalue and eigenvector, but our algorithm is capable of finding any number of eigenvalues/eigenvectors. We list the pre-scanning time as preprocessing time in tables. It only depends on the tensor dimension $n$ and unlike the sketching based method, it does not depend on $b$. Pre-scanning time is very short, because it only requires one pass of sequential access to the tensor which is very efficient on hardware.

**Sublinear time verification.** Our theoretical result suggests the total number of samples $b_{\text{no-prescan}}$ for our algorithm without pre-scanning is $n^{1-\alpha}(\alpha \in (0, 1])$ times larger than $b_{\text{prescan}}$ for our algorithm with pre-scanning. But in experiments we observe that when $b_{\text{no-prescan}} = b_{\text{prescan}}$ both algorithms achieve very similar accuracy, indicating that in practice $\alpha \approx 1$.

**Synthetic datasets.** We ran our algorithm on a large number of synthetic tensors with different dimensions and different eigengaps. Table 1 shows results for a tensor with 1200 dimensions with 100 non-zero eigenvalues decaying as $\lambda_i = \frac{1}{i^2}$. To reach roughly the same residual norm, the running time of our algorithm is over 50 times faster than that of the sketching-based robust tensor power method, thanks to the fact that we usually need a relatively small $B$ and $b$ to get a good residual, and the hidden constant factor in the running time of sampling is much smaller than that of sketching.

Our algorithm scales well on large tensors due to its sub-linear nature. In Figure 1(a), for the sketching-based method we kept $b = 2^{16}$, $B = 30$ for $n \leq 800$ and $B = 50$ for $n > 800$ (larger $n$ requires more sketches to observe a reasonable recovery). For our algorithm, we chose $b$ and $B$ such

that for each $n$, our residual norm is on-par or better than the sketching-based method. Our algorithm needs much less time than the sketching-based one over all dimensions. Another advantage of our algorithm is that there are zero or very minimal preprocessing steps. In Figure 1(b), we can see how the preprocessing time grows to prepare sketches when the dimension increases. For applications where only the first few eigenvectors are needed, the preprocessing time could be a large overhead.

**Real-life datasets** Due to the small tensor dimension (200), our algorithm shows less speedup than the sketching-based method. But it is still $2 \sim 6$ times faster in each of the six real-life datasets, achieving the same squared residual norm. Table 2 reports results for one of the datasets in many different settings of $(b, B)$. Like in synthetic datasets, we also empirically observe that the constant $b$ in importance sampling is much smaller than the $b$ used in sketching to get the same error guarantee.

| Sketching based robust power method: $n = \mathbf{1200}$, $\lambda_i = \frac{1}{i^2}$ | | | | | | | | | |
|---|---|---|---|---|---|---|---|---|---|
| | Squared residual norm | | | Running time (s) | | | Preprocessing time (s) | | |
| $b$ \ $B$ | 10 | 30 | 50 | 10 | 30 | 50 | 10 | 30 | 50 |
| $2^{10}$ | 1.010 | 1.014 | 0.5437 | 0.6114 | 2.423 | 4.374 | 5.361 | 15.85 | 26.08 |
| $2^{12}$ | 1.020 | 0.2271 | 0.1549 | 1.344 | 4.563 | 8.022 | 5.978 | 17.23 | 28.31 |
| $2^{14}$ | 0.1513 | 0.1097 | 0.1003 | 4.928 | 15.51 | 27.87 | 8.788 | 24.72 | 40.4 |
| $2^{16}$ | 0.1065 | 0.09242 | **0.08936** | 22.28 | 69.7 | **116.3** | 13.76 | 34.74 | **55.34** |
| Importance sampling based robust power method (without prescanning): $n = \mathbf{1200}$, $\lambda_i = \frac{1}{i^2}$ | | | | | | | | | |
| | Squared residual norm | | | Running time (s) | | | Preprocessing time (s) | | |
| $b$ \ $B$ | 10 | 30 | 50 | 10 | 30 | 50 | 10 | 30 | 50 |
| $5n$ | **0.08684** | 0.08637 | 0.08639 | **2.595** | 8.3 | 15.46 | **0.0** | 0.0 | 0.0 |
| $10n$ | 0.08784 | 0.08671 | 0.08627 | 4.42 | 13.68 | 25.84 | 0.0 | 0.0 | 0.0 |
| $20n$ | 0.08704 | 0.08700 | 0.08618 | 8.02 | 24.51 | 46.37 | 0.0 | 0.0 | 0.0 |
| $30n$ | 0.08697 | 0.08645 | 0.08625 | 11.63 | 35.35 | 66.71 | 0.0 | 0.0 | 0.0 |
| $40n$ | 0.08653 | 0.08664 | 0.08611 | 15.19 | 46.12 | 87.24 | 0.0 | 0.0 | 0.0 |
| Importance sampling based robust power method (with prescanning): $n = \mathbf{1200}$, $\lambda_i = \frac{1}{i^2}$ | | | | | | | | | |
| | Squared residual norm | | | Running time (s) | | | Preprocessing time (s) | | |
| $b$ \ $B$ | 10 | 30 | 50 | 10 | 30 | 50 | 10 | 30 | 50 |
| $5n$ | **0.08657** | 0.08684 | 0.08636 | **3.1** | 10.47 | 18 | **2.234** | 2.236 | 2.234 |
| $10n$ | 0.08741 | 0.08677 | 0.08668 | 5.427 | 17.43 | 30.26 | 2.232 | 2.233 | 2.233 |
| $20n$ | 0.08648 | 0.08624 | 0.08634 | 9.843 | 31.42 | 54.49 | 2.226 | 2.226 | 2.226 |
| $30n$ | 0.08635 | 0.08634 | 0.08615 | 14.33 | 45.4 | 63.85 | 2.226 | 2.224 | 2.227 |
| $40n$ | 0.08622 | 0.08652 | 0.08619 | 18.68 | 59.32 | 82.83 | 2.225 | 2.225 | 2.225 |

Table 1: Synthetic tensor decomposition using the robust tensor power method. We use an order-3 normalized dense tensor with dimension $n = 1200$ with $\sigma = 0.01$ noise added. We run sketching-based and sampling-based methods to find the first eigenvalue and eigenvector by setting $L = 50$, $T = 30$ and varying $B$ and $b$.

| Sketching based robust power method: dataset **wiki**, $\|\mathbf{T}\|_F^2 = 2.135\text{e+}07$ | | | | | |
|---|---|---|---|---|---|
| | Squared residual norm | | Running time (s) | | Preprocessing time (s) |
| $b$ \ $B$ | 10 | 30 | 10 | 30 | 10 | 30 |
| $2^{10}$ | 2.091e+07 | 1.951e+07 | 0.2346 | 0.8749 | 0.1727 | 0.2535 |
| $2^{11}$ | 1.971e+07 | 1.938e+07 | 0.4354 | 1.439 | 0.2408 | 0.3167 |
| $2^{12}$ | 1.947e+07 | 1.930e+07 | 1.035 | 2.912 | 0.4226 | 0.4275 |
| $2^{13}$ | **1.931e+07** | 1.927e+07 | **2.04** | 5.94 | **0.5783** | 0.6493 |
| $2^{14}$ | 1.928e+07 | 1.926e+07 | 4.577 | 13.93 | 1.045 | 1.121 |
| Importance sampling based robust power method (without prescanning): dataset **wiki**, $\|\mathbf{T}\|_F^2 = 2.135\text{e+}07$ | | | | | |
| | Squared residual norm | | Running time (s) | | Preprocessing time (s) |
| $b$ \ $B$ | 10 | 30 | 10 | 30 | 10 | 30 |
| $5n$ | **1.931e+07** | 1.928e+07 | **0.3698** | 1.146 | **0.0** | 0.0 |
| $10n$ | 1.931e+07 | 1.929e+07 | 0.5623 | 1.623 | 0.0 | 0.0 |
| $20n$ | 1.935e+07 | 1.926e+07 | 0.9767 | 2.729 | 0.0 | 0.0 |
| $30n$ | 1.929e+07 | 1.926e+07 | 1.286 | 3.699 | 0.0 | 0.0 |
| $40n$ | 1.928e+07 | 1.925e+07 | 1.692 | 4.552 | 0.0 | 0.0 |
| Importance sampling based robust power method (with prescanning): dataset **wiki**, $\|\mathbf{T}\|_F^2 = 2.135\text{e+}07$ | | | | | |
| | Squared residual norm | | Running time (s) | | Preprocessing time (s) |
| $b$ \ $B$ | 10 | 30 | 10 | 30 | 10 | 30 |
| $5n$ | **1.931e+07** | 1.930e+07 | **0.4376** | 1.168 | **0.01038** | 0.01103 |
| $10n$ | 1.928e+07 | 1.930e+07 | 0.6357 | 1.8 | 0.0104 | 0.01044 |
| $20n$ | 1.931e+07 | 1.927e+07 | 1.083 | 2.962 | 0.01102 | 0.01042 |
| $30n$ | 1.929e+07 | 1.925e+07 | 1.457 | 4.049 | 0.01102 | 0.01043 |
| $40n$ | 1.929e+07 | 1.925e+07 | 1.905 | 5.246 | 0.01105 | 0.01105 |

Table 2: Tensor decomposition in LDA on the wiki dataset. The tensor is generated by spectral LDA with dimension $200 \times 200 \times 200$. It is symmetric but not normalized. We fix $L = 50$, $T = 30$ and vary $B$ and $b$.

## Footnotes

[1]For two functions $f, g$, we use the shorthand $f \lesssim g$ (resp. $\gtrsim$) to indicate that $f \leq Cg$ (resp. $\geq$) for some absolute constant $C$.

[1] http://yining-wang.com/fftlda-code.zip

[2] https://github.com/huanzhang12/sampling_tensor_decomp/

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
