[Reviews · NeurIPS 2016]

Reviewer 1

Summary

Authors modify the robust power method to use approximate inner products computed via importance sampling. This results in analogous approximation guarantees to modifications which utilize sketching, but with more favorable computation when the tensor is in unstructured form. Experimental results support the analysis.

Qualitative Assessment

Authors clearly indicate the nature of the contribution: a reasonable idea (using importance sampling for inner product approximation instead of sketching) which has been apparently overlooked for tensor decomposition to date. The paper is thorough in exploring this idea both analytically and experimentally. Review scores reflect this reviewers impression as ``an extremely well executed, albeit incremental, contribution.''

Confidence in this Review

2-Confident (read it all; understood it all reasonably well)


Reviewer 2

Summary

The authors provide an approach towards construct a decomposition for an orthogonal tensor, in time sublinear in the size of the tensor. Their key insight is that the 'power iteration for tensors' involves multiplying the tensor with another tensor that has a compact implicit representation -- this allows us to estimate the product without reading the whole tensor. Given their algorithm for estimating this product, they appeal to the work of Wang et al. to show that such a noisy estimate still allows you to provably recover the tensor decomposition. The authors then present extensive experiments (synthetic and read -- tensors constructed from LDA models) to prove that this leads to practical speed ups over the previous result of Wang et al.

Qualitative Assessment

1. The core observation that the tensor products can be estimated quickly is insightful, and leads to immediate improvements in the existing algorithms for orthogonal tensor decomposition. The proofs are pretty straightforward. 2. The result about drawing m samples from a distribution on n items in O(m+n) time, though nice, is a very old result. See Alastair J. Walker. An efficient method for generating discrete random variables with general distributions. ACM Trans. Math. Softw., 3(3):253–256, September 1977. Or for a modern restatement of the result: see Karl Bringmann and Konstantinos Panagiotou. Efficient sampling methods for discrete distri- butions. In Automata, Languages, and Programming, pages 133–144. Springer, 2012. 3. There are several typos / minor issues in the paper: - Lemma 3 — It should be ‘with probability pqr’ not 1/pqr - Lemma 4 — Missing probability - Lemma 8 : The claim for hat(b) = nb requires more justification - Lemma 9 : In the variance expression, a 1/b factor is missing. - Theorem 10: Doesn’t the algorithm require O(nk) space? For storing the previous factors that have been computed. - Fig 1b : The scale on y-axis is unclear. 4. I like the key idea, and the speedup is very impressive in the initial experiments reported in the main paper. However, these speedups become less so in the other synthetic examples reported in the supplementary material (A factor of 2-3, which btw, is still impressive). However, for the real datasets, it is bothersome that somehow the residual error achieved using the sketching based approach seems to be a little better than the importance sampling based approach (this paper), though consistently, without a significant loss of running time. It would be very instructive if the authors could explain this behavior.

Confidence in this Review

2-Confident (read it all; understood it all reasonably well)


Reviewer 3

Summary

This paper proposes an sketching algorithm to compute tensor contraction via importance sampling, which is one per-iteration step for many orthogonal tensor decomposition algorithms like tensor power iteration, ALS. The authors show that importance sampling has the same (1) & (2) as FFT sketching. (1) concentration bound for tensor contraction (2) error bound for tensor power iteration

Qualitative Assessment

The main argument of this paper is that IS can achieve same accuracy must faster than FFT sketching, mainly because (i) can use much smaller sampling size than sketch size (ii) FFT has the overhead of sketching the whole tensor For (i), the authors provide some empirical evidence. Yet the theory established in this paper says the two will scale the same -- this suggest that the bound may be suboptimal. Besides, for all experiments on real datasets, the relative error is close to 1. I think it’s hard to assess their performance at this accuracy. How do they compare if we run them longer? For (ii), as they authors mentioned, if the tensor has factor form e.g. empirical moment tensor, the overhead of FFT can be significantly reduced. As before, because the theory shows the sampling and sketching size are same, the time complexity of FFT is actually better. This might lessen the potential impact of the proposed algorithm. To summarize, I think this paper give some empirical support for the effectiveness of IS, yet the theoretical side is not ready.

Confidence in this Review

2-Confident (read it all; understood it all reasonably well)


Reviewer 4

Summary

The authors present a novel strategy to efficiently compute the CANDECOMP/PARAFAC tensor decomposition. Their contribution is to provide a means to perform the necessary computation of the inner products in sublinear time with respect to the number of elements of the tensor to be decomposed making use of importance sampling. The importance sampling is in turn shown to be possible in linear time for a specified number of samples. The authors additionally show that the resulting estimator is unbiased and state its variance The runtime of the method is finally compared to a previously established algorithm that is linear in the number of non-zero elements of the tensor and found to be indeed significantly lower.

Qualitative Assessment

In my opinion, the overall quality of the paper is very high. The context and relevance as well as the contribution itself are clearly defined and thoroughly explained/proven. The experiments are reasonable and a comparision to the state of the art is provided. The results nicely confirm the theoretical findings. The only thing that one might miss is a brief summary or discussion at the very end. Please find below a list of further remarks. Languagewise, I just noticed a few minor errors, like in the first paragraph of the introduction ("A popular decomposition method is [the] canonical polyadic decomposition...") or in the last paragraph of the introduction ("from a theory standpoint[s]..."). In the function GENSORTEDRANDN, I believe the loops should run from 1 to m and not from 1 to L. Otherwise the stated complexity would not be correct. I however assume this to be only a typo. It should be stated in Figure 1 that the times are (I assume) given in seconds, like it is done in the tables. On page 6, there is a reference to Algorithm 5 when clearly Algorithm 4 is meant. You refer to Lemma 1 and 2 of reference 19, but there they are called Theorem 1 and 2. Furthermore, it seems that actually you only want to replace Theorem 1 and not 2, as Theorem 2 is in fact the main results that you claim carries over to the case of importance sampling.

Confidence in this Review

2-Confident (read it all; understood it all reasonably well)


Reviewer 5

Summary

This paper presents a randomized method for decomposing a symmetric, orthogonal tensor that is “nearly low rank”. Specifically, the tensor should expressible as a low-rank tensor + an arbitrary noise tensor with bounded norm. The fact that the tensor is symmetric and composed of orthogonal components may sound restrictive, but this is exactly the sort of tensor decomposition problem that comes up in spectral methods that solve Latent Dirichlet Allocation problems. So, it’s an interesting model to study. The method introduced very closely follows work in a paper from NIPS 2015, “Fast and guaranteed tensor decomposition via sketching” which uses a standard tensor power method for decomposition, but speeds up each iteration by accelerating the required tensor contractions (vector dot products against the tensor) via randomized sketching techniques. This paper uses the same algorithm, but instead of using sketches based on “random projections”, which take random linear combinations of all of the components of the tensor, it uses a subsampling technique to approximate the contractions. The authors main selling point is that this approach should allow for better running times since they don’t have to touch every entry in the tensor with each iteration. Sampling also avoids and expensive preprocessing cost incurred by the sketching algorithm. Without knowing the prior work, it’s a bit tough to understand how the runtime of this algorithm compares. In the introduction it’s stated that the sampling algorithm is slower per iteration than the sketching methods, which are accelerated with sparse hashing and fast Fourier methods. However, runtimes are stated in terms of the sketching dimension b which is not the same for both algorithms. It can vary by orders of magnitude in the experiments. It’s stated that the algorithm has a faster runtime overall, I believe because of preprocessing costs in the sketching algorithm, but I would like to see these costs laid out so that the overall runtime comparison is clear. Regardless, the algorithm performs well experimentally, providing a (3x-6x) speedup for small datasets and up to a 50x speedup for large synthetic data sets. This is especially impressive since the sampling algorithm is much simpler than the prior work, whose implementation seemingly requires many optimizations and detailed parameter setting.

Qualitative Assessment

This paper is decently written and the experimental results look promising. However, I don’t think the result is quite strong enough that I would recommend acceptance for NIPS. In general, the paper leans heavily on the work in “Fast and guaranteed tensor decomposition via sketching”. The idea to use sampling is nice, but the analysis is straightforward since the sampling scheme plugs into the prior work black-box. The authors do not prove any tighter theoretical guarantees and it’s not clear if their methods give any provable runtime improvements over the sketching methods. I would encourage the authors to clarify this aspect of their paper. More generally, while it’s nice to have theoretical guarantees, I worry that the bound in Lemma 10 is too complex/loose to provide indication of the algorithm’s quality. In addition to a very strong requirement on the noise eps, it has an inherent n^3 dependence, a stable rank dependence, a polynomial dependence on the failure probability, and spectral gap and condition number dependencies that I don’t expect to be small. It would be great to see if the analysis of these sketching techniques can be pushed beyond the work in “Fast and guaranteed tensor decomposition via sketching” to obtain a more interpretable/convincing bound. In light of it’s demonstrated practical performance, perhaps sampling can be analyzed in a tighter way? As a more specific point, I don’t understand the meaning of r(λ)=max(λ_i/λ_j). Wouldn’t this simply equal λ_1/λ_k? As far as presentation goes, I have a couple of comments: I think it’s a bit confusing to claim that the algorithm runs in “sublinear time”: it seems that the final runtime in Theorem 10 has a total O(n^3) dependence since b depends on n^2. I understand that that in practice the running time *can* be sublinear because the required number of samples is much lower than predicted by the theory, but I think this should be made more clear in the abstract and introduction. For a shortened form of the paper (like the NIPS submission) I would prefer to see the sampling analysis (currently in the supplemental material) in the main text, in lieu of the discussion of how to very efficiently generate importance samples. While I appreciate the careful work on this aspect of the algorithm (the actually sampling process is often overlooked by other papers!) ultimately it’s focusing on smallish log factors and is somewhat orthogonal to the main message of the paper. It could be relegated to an appendix.

Confidence in this Review

2-Confident (read it all; understood it all reasonably well)